# FT895 Impairs Mitochondrial Function in Malignant Peripheral Nerve Sheath Tumor Cells

**DOI:** 10.3390/ijms25010277

**Published:** 2023-12-24

**Authors:** Po-Yuan Huang, I-An Shih, Ying-Chih Liao, Huey-Ling You, Ming-Jen Lee

**Affiliations:** 1Department of Neurology, National Taiwan University Hospital, No. 7, Chung-Shan South Road, Taipei 10012, Taiwan; pyhuang0211@ntu.edu.tw (P.-Y.H.); 115338@ntuh.gov.tw (I.-A.S.); 122056@ntuh.gov.tw (Y.-C.L.); p94448012@ntu.edu.tw (H.-L.Y.); 2Department of Medical Genetics, National Taiwan University Hospital, Taipei 10012, Taiwan

**Keywords:** FT895, MPNST, mitochondrial function, XBP1s, GLUT1

## Abstract

Neurofibromatosis type 1 (NF1) stands as a prevalent neurocutaneous disorder. Approximately a quarter of NF1 patients experience the development of plexiform neurofibromas, potentially progressing into malignant peripheral nerve sheath tumors (MPNST). FT895, an HDAC11 inhibitor, exhibits potent anti-tumor effects on MPNST cells and enhances the cytotoxicity of cordycepin against MPNST. The study aims to investigate the molecular mechanism underlying FT895’s efficacy against MPNST cells. Initially, our study unveiled that FT895 disrupts mitochondrial biogenesis and function. Post-FT895 treatment, reactive oxygen species (ROS) in MPNST notably increased, while mitochondrial DNA copy numbers decreased significantly. Seahorse analysis indicated a considerable decrease in basal, maximal, and ATP-production-coupled respiration following FT895 treatment. Immunostaining highlighted FT895’s role in promoting mitochondrial aggregation without triggering mitophagy, possibly due to reduced levels of XBP1, Parkin, and PINK1 proteins. Moreover, the study using CHIP-qPCR analysis revealed a significant reduction in the copy numbers of promoters of the MPV17L2, POLG, TFAM, PINK1, and Parkin genes. The RNA-seq analysis underscored the prominent role of the HIF-1α signaling pathway post-FT895 treatment, aligning with the observed impairment in mitochondrial respiration. In summary, the study pioneers the revelation that FT895 induces mitochondrial respiratory damage in MPNST cells.

## 1. Introduction

Neurofibromatosis type 1 (NF1) is an autosomal dominant neurological disorder with a prevalence rate of 1 in 3500 [1]. NF1 is caused by mutations in the tumor-suppressive gene, *NF1*, which result in aberrant activation of the RAS signaling pathway [1]. The distinguishing features of NF1 include pigmentation of the skin and cutaneous neurofibromas that develop alongside peripheral nerves. These neurofibromas primarily consist of neoplastic Schwann cells, tissue matrix, and fibroblasts [2]. Approximately 25% of the NF1 patients develop one or more plexiform neurofibromas (PN) [3], which can transform into an aggressive sarcoma or malignant peripheral nerve sheath tumor (MPNST). NF1 patients have a 5–13% lifetime risk of developing an MPNST [3,4]; however, the relative risk is higher for those patients with a large plexiform neurofibroma [3,4]. MPNSTs exhibit significant resistance to chemotherapy and radiation therapies. Currently, complete surgical removal remains the sole effective treatment for MPNSTs. However, surgical excision is frequently constrained by factors such as the tumor’s dimensions, proximity to major blood vessels, adjacency to vital organs, or the potential for early metastatic spread. The frequent instances of recurrence and metastasis contribute to a notably low five-year survival rate among individuals with NF1-associated MPNSTs [5,6].

FT895 is a histone deacetylase 11 (HDAC11)-specific inhibitor with an IC_50_ of 3 nM. It showed more than 1000-fold inhibitory activity for HDAC11 over other histone deacetylases (HDACs) [7]. Several studies have demonstrated the potential of using FT895 to reduce the proliferation of tumor cells in vitro. For example, FT895 effectively hindered the proliferation of JAK2-driven myeloproliferative neoplasms through the inhibition of the JAK/STAT signaling pathway [8]. Another noteworthy finding is the FT895’s ability to diminish the self-renewal capacity of cancer stem cells from non-small-cell lung cancer. It effectively restrains the expansion of drug-resistant cancer stem cells by down-regulating the expression of Sox2, an essential transcription factor for maintaining pluripotency. Additionally, FT895 exerts a suppressive effect on the stemness of lung adenocarcinoma cells and decreases the viability of cells resistant to EGFR tyrosine kinase inhibitors (TKIs) [9]. We recently showed a synergistic effect between cordycepin and FT895 in suppressing the proliferation of MPNST cells both in vitro and in vivo [10]. This study also revealed that the increase in MPNST cells’ cordycepin sensitivity is mediated through the disruption of the Hippo signaling pathway by reducing the expression of key transcription factor TEAD1 (TEA domain transcription factor 1) and its associated protein, TAZ (TAFAZZIN, Tafazzin, phospholipid-lysophospholipid transacylase) [10]. However, the mechanisms responsible for the functional disturbance of MPNST cells following FT895 treatment remain unclear. Aside from its HDAC11 inhibitory activity, this study aims to investigate the impact of FT895 treatment on MPNST cells, particularly focusing on mitochondrial dysfunction since TAZ is involved in the synthesis of phospholipid cardiolipin, a key component of the inner mitochondrial membrane. Our findings suggest that FT895 potentially influences mitochondrial functions, resulting in a reduction in the expression of proteins associated with both its function and biogenesis. This is the first time we have demonstrated the impact of FT895 on mitochondria.

## 2. Results

### 2.1. Treatment of FT895 Induced the Production of Reactive Oxygen Species in MPNST Cells

When cells were treated with HDAC inhibitors, the balance of reactive oxygen species (ROS) and antioxidants in cells was disrupted [11]. Oxidative stress can lead to cell death when the generation of ROS overwhelms the antioxidant defenses in cells. FT895 is a selective HDAC inhibitor; therefore, the initial focus to uncover the mechanism underlying FT895’s synergistic effect on the cytotoxicity of cordycepin in MPNST cells is to examine the generation of ROS. One of the most straightforward methods to measure ROS is to use the permeable 2′-7′-dichlorodihydrofluorescein diacetate (DCFDA) with flowcytometry analysis. Upon subjecting four different MPNST cell lines to a 24 h treatment of 10 μM FT895, a discernible shift in the 2′-7′-DCFDA peak denoting the cells with accumulated ROS became evident (Figure 1a). Further analysis to determine the mean fluorescent intensity revealed a significant increase in ROS levels in S462TY, STS26T, and T265 cells, while a modest elevation of ROS in ST8814 cells after treatment (Figure 1b). To confirm whether the rise in ROS levels resulted from HDAC11 inhibition, we administered SIS17, a different selective HDAC11 inhibitor, to S462TY and STS26T. SIS17 exhibits a strong binding affinity to HDAC11, with an IC50 value of 0.83 µM, similar to FT895. Notably, it does not interfere with the activity of other HDACs, as demonstrated by Son et al. [12]. Following 24 h treatment with 10 µM SIS17, we assessed the intracellular ROS levels. The FACS data revealed no discernible shift in the SIS17-treated S462TY and STS265T cells (Figure 1c). Moreover, there is no significant difference in the mean fluorescent intensities of SIS17 in S463TY and STS26T cells compared to those treated with DMSO (Figure 1d). The acetylated histone H3 (Ac-H3) levels in MPNST cells after treatment with 10 μM of FT895 were also investigated; however, FT895 did not significantly increase the Ac-H3 level in the four MPNST cell lines (Figure 1e). These results indicate that the rise in ROS levels after FT895 treatment might not be attributed to HDAC11 inhibition. Instead, it is plausible that the functional alterations in mitochondria could contribute to the substantial increase in ROS production following treatment. The quantity of mitochondrial DNA (mtDNA-CN) holds great potential as a biomarker for mitochondrial dysfunction [13]. We employed primer sets specifically designed to target the mitochondrial genes ND2 and ND4L regions on the mitochondrial chromosome. Following a 24 h exposure to 10 μM FT895, the levels of mtDNA-CN based on both ND2 and ND4L genes exhibited a significant reduction in all four different MPNST cells (Figure 1f). These findings imply a potential impact on the mitochondrial function of the MPNST cells after FT895 treatment.

### 2.2. FT895 Interferes with the Mitochondrial Respiration

After administering the FT895 treatment, there was a notable reduction in mitochondrial DNA copies, suggesting a potential influence on mitochondrial activity and generation. To further explore the impact of FT895 on mitochondrial function, we conducted a direct assessment of mitochondrial respiration activities using the Mito Stress Test. Following a 24 h exposure to 10 μM FT895, we measured the oxygen consumption rate (OCR). As shown in Figure 2a, kinetic measurements showed the reduced activities of mitochondrial respiration when FT895 was administered to S462TY, T265, and ST8814 cell lines, while STS26T showed a modest alteration. Furthermore, S462TY and T265 displayed no reaction to the treatment involving an ATP synthase inhibitor, oxidative phosphorylation uncoupler, and complex I/III inhibitors, whereas STS26T and ST8814 exhibited only a partial response. Data analysis found that, in comparison to the DMSO group, the FT895 group exhibited a significant decrease in basal respiration, maximal respiration, and ATP-production-coupled respiration in four cell lines (Figure 2b–d). When considering the data on extracellular acidification (ECAR), which reflects the acidification of the culture media due to lactate produced from glycolysis, we noted that FT895 induced a relatively less active state in S462TY and T265 cells, whereas STS26T cells became more energetically active (Figure 2e). Furthermore, STS8814 cells appeared to show a preference for the glycolytic pathway as their energy source following FT895 treatment. These findings imply that S462TY and T265 cells exhibit a higher sensitivity to the mitochondrial inhibition caused by FT895 compared to STS26T and ST8814 cells. Given that the OCR remained consistently low in response to 24 h FT895 treatment in S462TY and T265, we examined the impact of a shorter FT895 treatment duration on mitochondria in these cell lines. A six-hour exposure to 10 and 20 μM FT895 suppressed mitochondrial respiration in a dose-dependent manner, resulting in significant reductions in both basal and maximal respiration (Figure 2f–h). These findings indicate that FT895 influences mitochondrial function as early as 6 h after treatment, and the effect continues at least for 24 h.

### 2.3. FT895 Promotes Mitochondrial Aggregation without Triggering Mitophagy

To examine both the cell’s characteristics and organelle integrity, we used forward scatter (FSC) and side scatter (SSC) from flower cytometry to examine the impact of FT895 on cells. The FSC signal can distinguish cells based on their size, whereas the SSC signal arises from the interaction of light with intracellular structures such as granules and the nucleus, providing insights into the internal complexity or granularity of the cells. This granularity information is particularly useful for detecting any alterations in mitochondria following the treatment. While analyzing FT895-treated MPNST cells using FACS, we observed an upward shift in side scatter (SSC), particularly in S462TY cells, indicating an increase in cellular granularity (see Figure 3a). Mitophagy is the process responsible for removing damaged mitochondria from cells, involving the sequestration of mitochondria by autophagosomes. The presence of an elevated number of autophagosomes within cells can lead to light deflection, resulting in a high SSC signal on FACS. To explore the possible involvement of autophagosomes in S462TY after FT895 treatment, we labeled mitochondrial ATP synthase ATP5H (ATP5PD, ATP synthase peripheral stalk subunit D) and the autophagosome marker LC3B (MAP1LC3B, microtubule associated protein 1 light chain 3 beta) in S462TY cells and tracked the localization of mitochondria and autophagosomes. There was no noticeable co-localization of LC3B with mitochondria following FT895 treatment, indicating that abnormal mitophagy was not the cause of the defective mitochondria (Figure 3b). Instead, we observed aggregated mitochondria positioned around the nucleus periphery after FT895 treatment. This finding explains the upward shift in SSC and aligns with the previous report suggesting that apoptotic stimulation induces mitochondrial aggregation in fibrosarcoma cells [14].

### 2.4. FT895 Impairs the Mitochondrial Dynamic Pathway

To further study the mechanism of the disruption of mitochondrial function without triggering mitophagy by FT895, we examined the expression of PINK1 (PTEN-induced kinase 1) and Parkin (PRKN, Parkin RBR E3 ubiquitin protein ligase) in FT895-treated MPNST cells. Our experiment revealed a decrease in PINK1 and Parkin protein levels in S462TY, STS26T, T265, and STS8814 cells following FT895 treatment (Figure 4a). These results imply that FT895 compromises the PINK1-Parkin pathway, potentially contributing to the deficiency in mitophagy.

X-box binding protein 1 (XBP1) is a transcription factor that undergoes activation when there is endoplasmic reticulum (ER) stress. This splicing event results in the formation of a spliced variant, XBP1s [15]. The XBP1s exerts control over mitophagy by regulating PINK1 expression at the transcriptional level. Additionally, PINK1-dependent phosphorylation of XBP1s influences its transcriptional activity, leading to its translocation into the nucleus [16]. Given the crucial interplay between PINK1 and XBP1s in governing mitophagy, we then evaluated the expression of XBP1 in MPNST cells treated with FT895. Notably, the mRNA sequences of unspliced (XBP1u) and spliced (XBP1s) XBP1 mRNA differ by 26 nucleotides at the 5′ region, making it challenging to distinguish them accurately using quantitative PCR. However, the removal of these 26 nucleotides induces a frameshift in XBP1s, resulting in distinct amino acid sequences compared to XBP1u. To examine the XBP1 proteins in FT895-treated MPNST cells, we employed anti-XBP1u (unspliced XBP1) and anti-XBP1s antibodies. The results revealed a constant XBP1u level in MPNST cells and a decrease in the XBP1s level in S462TY, T265, and ST8814 cells after FT895 treatment (Figure 4b). This finding corroborates the notion that the disruption of the XBP1s-PINK1-Parkin pathway may contribute to the lack of mitophagy in FT895-treated cells.

### 2.5. XBP1s Regulates Mitochondria-Related Genes at the Transcriptional Level

XBP1, a transcription factor, is known to modulate a range of cellular processes, including the management of endoplasmic reticulum (ER) stress, immune responses, and the progression of tumors [15,17]. We further explored whether XBP1 can epigenetically modify the transcription of genes related to mitochondria. It has been found that the promoters’ binding sites for XBP1s include the ACGT core and CAAT box motifs [18]. Herein, we selected the candidate genes MPV17L2, POLG, TFAM, PGC1α, PINK1, and Parkin, which are related to mitochondrial function and biogenesis, as well as the ACGT core and CAAT box motifs in the promoter region (Figure 5a). MPV17L2 (MPV17 mitochondrial inner membrane protein like 2) is a protein located in the inner mitochondrial membrane that interacts with the mitochondrial ribosome and facilitates mitochondrial translation [19]. POLG (DNA polymerase gamma, catalytic subunit) operates as the catalytic subunit of mitochondrial DNA polymerase, responsible for the replication of the mitochondrial genome [20]. TFAM (transcription factor A, mitochondrial) is a protein encoded in the nucleus that oversees mitochondrial transcription and the quantity of mitochondrial DNA copies [21]. PGC1α (PPARGC1A, PPARG coactivator 1 alpha), a transcriptional co-activator, is recognized as a key regulator in the transcriptional control of mitochondrial biogenesis and quality [22].

The protein levels of XBP1s decreased in MPNST cells after FT895 treatment (Figure 4b). To investigate the possible XBP1s-mediated transcriptional control on MPV17L2, POLG, TFAM, PGC1α, PINK1, and Parkin genes, a chromatin immunoprecipitation (ChIP) assay with an XBP1s antibody was employed. This allowed us to capture DNA fragments bound to XBP1s, which were subsequently quantified through qPCR with primers that are specific for the promoter regions of the genes listed above. As shown in Figure 5b, the XBP1s-mediated transcriptional activities for MPV17L2, POLG, TFAM, PINK1, and Parkin notably decreased in S462TY, STS26T, and T265 cells after FT895 treatment but had no effect in ST8814 cells. Among these cell lines, only S462TY displayed reduced transcriptional activity for PGC1α (Figure 5b). The mRNA expression levels of Parkin, PINK1, PGC1α, POLG, and MPV17L2 were evaluated by quantitative real-time PCR. The levels of Parkin, PINK1, POLG, and MPV17L2 were significantly decreased in all four MPNST cell lines, including ST8814 (Figure 5c). Concerning the differential expression in ST8814, it might be a cell-line-specific phenomenon. Since several transcription factors are responsible for the regulation of PINK1 and Parkin expression, e.g., NRF2, NRF2, and NFκB [23,24,25], the reduction in PINK1 and Parkin expression might be achieved by different mechanisms in ST8814 with the FT895 treatment. There is no significant difference in the expression levels of PCG1α in T265 and ST8814 cells after FT895 treatment, albeit a significant decrease in the levels in S462TY and STS26T cells (Figure 5c). The reduction in promoter DNA sequences signifies a decrease in transcriptional efficiency, which may lead to impaired mitochondrial metabolism and biogenesis in the MPNST cells following FT895 treatment. The decrease in PINK1 and Parkin protein levels aligns with these findings as well (Figure 4a).

### 2.6. RNA-Seq Reveals the Gene Expression Affected by FT895 in MPNST Cells

To assess the impact of FT895, we employed RNA-seq to elucidate its effects on the transcriptome. Here, we specifically selected S462TY cells due to their sensitivity to FT895. To obtain early-stage gene expression changes, the 6 h FT895 treatment was selected. The gene set enrichment analysis with ClusterProfiler demonstrated that among the top 10 KEGG pathways affected, the HIF-1α signaling pathway featured prominently, which aligns with our observation of FT895-induced mitochondrial respiration impairment (see Figure 6a). Furthermore, the enrichment analysis demonstrated that FT895 treatment activated pathways associated with carbohydrate metabolism and tumor survival signaling, including fructose and mannose metabolism, as well as breast cancer-related signaling (see Figure 6a). Detailed information regarding these pathways can be found in Table 1.

Using the STRING database to analyze the functional protein association networks, we identified SLC2A1, the gene encoding glucose transporter 1 (GLUT1), as the central node in the network (see Figure 6b). Previous studies have established that activation of the HIF-α signaling pathway leads to an increase in GLUT1 expression [26]. Further investigation of the mRNA expression of GLUT1 in MPNST cells showed increased expression in S462TY, T265, and ST8814 but not in STS26T cells after treating FT895 (Figure 7a). The changes in GLUT1 protein levels in MPNST cells were consistent with the mRNA expression with FT895 treatment (Figure 7b). These findings suggest that genes related to carbohydrate metabolism were up-regulated in response to the loss of mitochondrial function after FT895 treatment.

## 3. Discussion

The lack of effective treatment for MPNSTs poses a life-long threat to NF1 patients. FT895, an HDAC11 inhibitor, was previously found to enhance the cytotoxicity of cordycepin in MPNST cells and suppress MPNST cell growth both in vitro and in vivo. Treating MPNST cells with FT895 alone also induces a certain degree of cell death, but the molecular impact of FT895 on MPNST cells is not fully understood. This study demonstrated that FT895 led to an increase in ROS production in MPNST cells (Figure 1a,b). Mitochondria are the primary source of ROS in mammalian cells, and excessive ROS generation frequently results from aberrant mitochondrial function. Our finding of reduced mitochondrial DNA copy numbers in MPNST cells indicated impaired mitochondrial activities after FT895 treatment (Figure 1f). In addition, mitochondrial respiration measurements also revealed an overall decrease in FT895-treated cells. This included decreased basal respiration, maximal respiration, and ATP production-coupled respiration in FT895-treated cells (Figure 2). Interestingly, mitochondrial aggregation was observed in FT895-treated MPNST cells; however, this phenomenon seems not to induce mitophagy. With further investigation of the PINK1-Parkin pathway, which is crucial for mitophagy, FT895 treatment was found to decrease the levels of PINK1 and Parkin proteins in MPNST cells, which may impede mitophagy. The spliced variant XBP1, XBP1s, a transcription factor that regulates PINK1 expression, was also decreased in FT895-treated MPNST cells (Figure 4b). The concentrations of several genes related to mitochondrial functions, such as PINK1, Parkin, POLG, MPV17L2, PGC1α, and TFAM, that harbor the binding sequences of XBP1s in their promoter regions were also affected. Therefore, the decrease in XBP1s may down-regulate mitochondrial-related genes, which subsequently impaired mitophagy, as illustrated in Figure 8. RNA-seq analysis of FT895-treated S462TY cells revealed changes in the transcriptome, particularly in the genes involved in pathways related to HIF-1α signaling, carbohydrate metabolism, as well as the tumor survival pathway. GLUT1, a key glucose transporter in the blood–brain barrier, was identified as the central node in the affected biological processes, suggesting an up-regulated carbohydrate metabolism that may be involved in compensating mitochondrial dysfunctions induced by FT895. Finally, in the cells treated with FT895, validation by qPCR and immunoblotting confirmed an increase in both mRNA and protein levels of GLUT1.

HDAC11 is a histone deacetylase with significant long-chain fatty acid deacetylase activity [27,28]. HDAC11 may also be involved in mitochondrial function through the promotion of the glycolytic-to-oxidative muscle fiber switch, leading to an increased number of oxidative myofibers, which can improve muscle strength and fatigue resistance [29]. The absence of HDAC11 led to an increase in mitochondrial content, manifested by higher mtDNA copy numbers and increased expression of the mitochondrial transcription factor TFAM [29]. Although the study highlighted the possible role of HDAC11 in the up-regulation of mitochondrial function, it has not explored its pathological involvement, such as in cancers. Recently, we reported that FT895, a specific HDAC11 inhibitor, could synergistically enhance the cytotoxicity of cordycepin in MPNST cells, especially in the less cordycepin-sensitive cell line, S462TY. FT895 alone could also suppress the growth of MPNST cells in vitro and *in vivo*. We found that the synergistic effect depended on the hippo signaling pathway [10]. Nevertheless, the specific molecular pathway or organelle damage responsible for the anti-tumor effect of FT895 in MPNST cells remains unknown. In the current study, we showed that treating MPNST cells with FT895 led to decreased mitochondrial content, damaged mitochondrial respiration, and cell death. Notably, the treatment of SIS17, another HDAC11 inhibitor [12], did not induce either ROS production in MPNST cells or significant cell death (Figure 1c). These results suggest the impact of FT895 on MPNST cells may not be mediated exclusively through HDAC11 inhibition but rather by the interruption of mitochondrial homeostasis. Further research is necessary for us to understand the role of HDAC11 in mitochondrial function.

The primary origin of ROS within mammalian cells arises from the mitochondria during the process of respiration [30]. The surplus generation of ROS when respiration is hindered was found to be associated with the depletion of NAD(+)-linked substrates oxidized by cytochrome c oxidase. According to Kushnareva et al., the production of ROS within mitochondria occurs close to the inhibitory site of rotenone, specifically at complex I, rather than at complex III [31]. After the treatment with FT895, the mitochondrial DNA copy number measured based on the ND2 and ND4L DNA regions decreased significantly. The mitochondrial genes ND2 and ND4L encode proteins that are part of complex I, also known as NADH: Ubiquinone oxidoreductase, in the mitochondrial electron transport chain (ETC). The large protein complex is embedded in the mitochondrial inner membrane, which transfers electrons from NADH to ubiquinone (coenzyme Q) in the ETC. Complex I plays a central role in the process of oxidative phosphorylation to generate ATP. During the process, a small fraction of electrons can “leak” from the ETC, which may react with oxygen (O_2_) to form superoxide radicals (O_2_•−) [32]. The production of superoxide radicals represents a natural by-product of mitochondrial respiration, which is highly reactive and can damage cellular components such as proteins, lipids, and DNA. In physiological conditions, the superoxides can be neutralized by superoxide dismutase (SOD) and glutathione peroxidase. An impairment of the ETC process in complex I can result in elevated redoxed oxygen molecules followed by superoxide overproduction, which may further damage the mitochondria. The increase in oxidative stress after FT895 treatment impacted the expression of proteins that are vital for electron transport, thus facilitating a detrimental feedback loop of ROS, which further aggravates the organelle dysregulation that ultimately culminates in apoptosis [33].

Removing damaged mitochondria through mitophagy, a specialized autophagic machinery, is essential for maintaining mitochondrial homeostasis. PINK1 (PTEN-induced kinase 1) is an upstream regulator of Parkin’s function by facilitating the translocation of Parkin, an E3 ubiquitin ligase, from the cytosol to damaged mitochondria, initiating mitophagy to eliminate these damaged mitochondria [34]. The ignition of mitophagy involves the translocation of PINK1 to the mitochondrial outer membrane, subsequently recruiting Parkin to the damaged mitochondria. Parkin will then ubiquitinate numerous proteins on the outer mitochondrial membrane, which promotes the recruitment of ubiquitin-binding autophagy receptors such as p62 to mark the damaged mitochondria to LC3-positive phagophores for clearance in the lysosome [34,35]. The damaged mitochondria are targeted and engulfed into autophagosomes for protein degradation and recycling. Mitophagy can prevent excessive ROS production from the damaged mitochondria; thus, PINK1 and Parkin play essential roles in mitochondrial homeostasis. The treatment of FT895 significantly reduced the levels of both proteins, which may have implications for the FT895-mediated cell death process.

In addition to the reduced levels of PINK1 and Parkin proteins, a decrease in XBP1s levels was also found in the FT895-treated MPNST cells. XBP1 is a transcription factor involved in the unfolded protein response (UPR), which is activated in response to endoplasmic reticulum (ER) stress. Physiologically, the UPR aims to restore ER homeostasis by increasing the production of chaperones and enzymes involved in protein folding, as well as reducing the load of protein synthesis [36]. There is intricate communication between the ER and mitochondria within cells, and disruptions in ER homeostasis result in mitochondrial dysfunction and mitochondrial stress [37]. It has been reported that upon ER stress, the endonuclease activity of IRE1 (inositol-requiring enzyme 1) mediates the splicing of XBP1 mRNA to generate the spliced form, XBP1s [15]. XBP1s translocates from the cytoplasm to the nucleus and regulates the expression of various genes involved in ER homeostasis and the UPR [38]. Our findings showed that XBP1s also affect the levels of genes involved in the biogenesis and degradation of mitochondria, such as POLG1, MPV17, PINK1, and Parkin, etc. (Figure 5). Recent studies have demonstrated that XBP1s appears to have a role in regulating mitophagy [16]. By promoting mitophagy, XBP1s helps eliminate compromised mitochondria and mitigate oxidative stress, contributing to overall cellular health. Thus, XBP1/XBP1s, a known player in the UPR during ER stress, is increasingly recognized as an important part of mitophagy and mitochondrial homeostasis. This relationship highlights the capability of cells to coordinate the responses to stress in different organelles, ultimately aiming to maintain cellular function and prevent further effects on organelle function. With the stress induced by FT895 treatment, the significant down-regulation of XBP1s may be one of the culprits leading to the defect in mitophagy, which furthers the production of ROS.

Like most malignant cells, MPNST cells tend to use glycolysis to generate energy [39]. However, the haploinsufficiency of the neurofibromin in MPNST cells may lead to a reduction in mitochondrial respiratory function and glycolytic rates compared to non-NF1 cells [39,40]. STS26T, the non-NF1-derived MPNST cells, showed more energy in glycolysis and mitochondrial respiratory rate compared to the NF1-derived MPNST cells [39]. It suggested that neurofibromin might be involved in the regulation of energy metabolism. The metabolism of S462TY cells with the treatment of FT895 shifted toward a more energetic state, which might be due to the inadequacy of neurofibromin. Recently, Allaway et al. reported that a synthetic small molecule Y100 could induce NF1-deficient MPNST cell death through the disruption of metabolic homeostasis and induce the formation of mitochondrial superoxide [41]. In addition, NF1 dysregulation has been shown to increase sensitivity to oxidative stress induced by HSP90 inhibitors [42]. The available data, including findings from this report, indicate mitochondrial function might be a treatment target in NF1-deficient MPNSTs. However, additional research is needed to comprehend why carbohydrate metabolism, serving as the primary energy source, cannot compensate for mitochondrial dysfunction.

## 4. Materials and Methods

### 4.1. Cell Culture

The S462TY cell line was a kind gift from Dr. Timothy Cripe (Nationwide Children’s Hospital Columbus, Columbus, OH, USA) [43]. The MPNST cell lines STS26T, T265, and ST8814 have been reported before [44] and were gifted by Dr. Nancy Ratner (Cincinnati Children’s Hospital, Cincinnati, OH, USA). The cultures were supplemented with DMEM and 10% fetal bovine serum in a humid incubator at 37 °C with 5% CO_2_. All MPNST cell lines, except STS26T, were derived from NF1 patients. STS26T is a sporadic MPNST with no known NF1 mutation [45]. Normal human Schwann cells (HSC) (ScienCell Research Laboratories, Carlsbad, CA, USA) were grown on the poly-L-lysine-coated tissue culture plates.

### 4.2. Reactive Oxygen Species Detection

To identify the intracellular reactive oxygen species (ROS), a cell-permeable fluorescent probe, oxidized 2′,7′-dichlorofluorescein diacetate (DCFDA) (Sigma-Aldrich, Taipei, Taiwan), was used as a fluorescent ROS probe. MPNST cells treated with DMSO or 10 μM of FT895 for 24 h were loaded with 1 μM DCFDA and incubated for 20 min. The fluorescent signal of oxidized DCFDA was excited by a 488 nm laser and detected by the FITC channel in FACSverse Flow Cytometry (BD Biosciences, Franklin Lakes, NJ, USA). The data were analyzed by v10.8 of FlowJo™ software.

### 4.3. Mitochondrial DNA Copy Number

Genomic DNA was extracted using the Genomic DNA Kit (Geneaid Biotech Ltd., Taoyuan, Taiwan) and quantified with SpectraMax^®^ i3x (Molecular Devices, San Jose, CA, USA). The relative DNA copy number (mitochondrial DNA to nuclear DNA) of three mitochondrial genes, ND2, ND4L, and nuclear GAPDH, was measured by quantitative real-time PCR (qPCR). The primer sets are listed as follows: ND2 forward: CATATACCAAATCTCTCCCTC, ND2 reverse: GTGCGAGATAGTAGTAGGGTC; ND4L forward: TAGTATATCGCTCACACCTC, ND4L reverse: GTAGTCTAGGCCATATGTG; GAPDH forward: GAAGGTGAAGGTCGGAGTC, GAPDH reverse: GAAGATGGTGATGGGATTTC. The mitochondrial copy number (mtCN) was quantified using the delta CT (ΔCT) of mtDNA and nuclear DNA (ΔCt = CT [mtDNA] − CT [GAPDH]). The fold change was calculated by 2^−ΔΔCT^. ΔΔCT was determined by ΔCT [FT895 treated] − ΔCT [DMSO control].

### 4.4. Metabolism Assay

MPNST cells were plated in a Seahorse XFe24 microplate (Agilent Technologies, Santa Clara, CA, USA) and allowed to adhere overnight. An XFe24 FluxPack sensor cartridge (Agilent Technologies, Santa Clara, CA, USA) is equilibrated in XF Calibrant at 37 °C overnight. Before performing the assay, cells were washed twice in a standard XF DMEM medium supplemented with pyruvate, glutamine, and glucose. Cells were then incubated in 675 μL of XF DMEM media for one hour at 37 °C with atmospheric CO_2_. The Mito Stress assay was run as described in the test kit protocol (Agilent Technologies, Santa Clara, CA, USA) with 1 μM oligomycin and 0.5 μM rotenone/antimycin A. The concentration of carbonyl cyanide-4 (trifluoromethoxy)-phenylhydrazone (FCCP) for each MPNST cell line was 1 μM for S462TY, 1.5 μM for STS26T, and 2 μM for T265 and ST8814 cells. The data were collected to determine the changes in oxygen consumption rate (OCR) and extracellular acidification rate (ECAR). The analysis was performed with Seahorse Wave Pro software (Agilent Technologies, Santa Clara, CA, USA, https://www.agilent.com.cn/zh-cn/product/cell-analysis/real-time-cell-metabolic-analysis/xf-software/software-download-for-seahorse-wave-pro-software, accessed on 19 December 2023).

### 4.5. Chromatin Immunoprecipitation (ChIP) Assay and Quantitative PCR

The ChIP assay was performed following the manufacturer’s instructions for the Pierce Magnetic ChIP Kit (Thermo Fisher Scientific, Waltham, MA, USA). In brief, the chromatin extracted from the control and treated cells was digested with MNase and sheared by sonication to a size of less than 400 bp. The fragments were centrifuged (9000× *g*) at 4 °C for 5 min. Ten percent of the diluted supernatant (input) was collected and served as the control template. The sheared chromatin was immunoprecipitated with 5 μg of anti-XBP1 antibodies (Cell Signaling Technology, Danvers, MA, USA) overnight. Magnetic beads and a magnetic separation device (Merck, Darmstadt, Germany) were employed to pull down the protein/DNA complexes from the crude chromatin mixture. After elution, the DNA from immunoprecipitated samples and the inputs were recovered by phenol/chloroform. The input DNA and antibody-bound chromatin DNA were then subjected to quantitative PCR to quantify the DNA copy number. The primers to amplify the promoter region of the *POLG*, *MPV17L2*, *NOTCH2*, *PINK1*, *PARK2* (encodes Parkin), *PPARGC1A* (encodes PGC1α), and *TFAM* genes were designed. The primer sequences are as follows: *POLG* forward: CAATCT CGCAGGGACTTGCT, *POLG* reverse: CAATGTGAACGTAGTCGCCG; *MPV17L2* forward: CCCATAGGCTGGCACAGAAA, *MPV17L2* reverse: GGCGTTCTATCCCAAGGTCG; *PINK1* forward: GGCAAGCTGCTATCTTGGGA, *PINK1* reverse: CTCAATGCCGTTAGGCTGCT; *PARK2* forward: CCTTGGCTAGAGCTGCAACA, *PARK2* reverse: TCAGGCCCAGCAATCTTACG; *PPARGC1A* forward: CGTCACGAGTTAGAGCAGCA, *PPARGC1A* reverse: TCCCCAGTCACATGACAAAGC; *TFAM* forward: TTAGGTTTGCGAATCCCCGC, *TFAM* reverse: CGTAACAGACAGTCCTGCATCC. The primer set used to quantify the mRNA level of *SLC2A1* encoding GLUT1 and housekeeping gene *GAPDH* are as follows: *SLC2A1* forward: CTGGCATCAACGCTGTCTTC, *SLC2A1* reverse: GTTGACGATACCGGAGCCAA; *GAPDH* forward: ACCCACTCCTCCACCTTTGA, *GAPDH* reverse: CTGTTGCTGTAGCCAAATTCGT.

### 4.6. Western Blot

Equal amounts of the protein lysate from control and treated cells were separated by SDS-PAGE, followed by transfer to a PVDF membrane. After blocking with 5% skim milk for one hour at room temperature, the membrane was probed with the specific primary antibody overnight at 4 °C. The primary antibodies used in the study include anti-GAPDH (Proteintech Group Inc., Rosemont, IL, USA), anti-α-tubulin (Proteintech Group Inc., Rosemont, IL, USA), anti-XBP1u (Abcepta, Reservoir Victoria, Australia), anti-XBP1s (Cell Signaling), anti-Parkin (Cell Signaling, Danvers, MA, USA), anti-PINK1 (Genetex, Hsinchu, Taiwan), anti-LC3B (Cell Signaling, Danvers, MA, USA), and anti-GLUT1 (Proteintech Group Inc., Rosemont, IL, USA) antibodies. Then, the membranes were washed with PBS containing 0.1% Tween 20 to remove unbound antibodies. The membrane was then incubated with horseradish peroxidase-conjugated secondary antibody for two hours. The protein signals were detected using the enhanced chemiluminescence method, and the snapshot images were captured by the UVP BioSpectrum 810 Imaging System (UVP, Cambridge, UK).

### 4.7. Immunofluorescence and Confocal Microscopy

Cells were grown on Ibid tissue culture dishes (IB-81156, Ibidi, Gräfelfing, Germany) overnight and were treated with FT895 for 24 h. Treated cells were washed with PBS twice and fixed in 4% paraformaldehyde at room temperature (RT) for one minute, followed by fixing with cold methanol at −20 °C for 20 min. The fixed cells were blocked with PBS containing 5% fetal bovine serum for one hour at RT. For immunostaining, the primary antibodies, anti-ATP5H (Thermo Fisher Scientific, Waltham, MA, USA), and anti-LC3 antibodies (Cell Signaling, Danvers, MA, USA), were added into the wells and incubated overnight at 4 °C. The cell nuclei were stained with DAPI for 10 min at RT, and the cells were mounted using a mounting medium for microscopy. The confocal images were acquired with a Zeiss LSM 880 (Zeiss, Oberkochen, Germany).

### 4.8. Bulk RNA-Seq Analysis

Total RNAs were isolated from the S462TY cells both with and without 6 h treatment with 10 µM FT895 by the phenol/chloroform extraction method. RNA libraries were prepared by the Illumina Truseq Stranded mRNA Kit. RNA sequencing was carried out on a NovaSeq 6000 system, and the read quality was checked with FASTQC. To align the reads, we employed HISAT2 version 2.2.1 [46] against the GRCh38 human genome. Read counts were normalized by StringTie version 2.2.1 [47]. Differential expression analysis was analyzed by DEGseq version 1.48.0 (https://bioconductor.org/packages/devel/bioc/manuals/DEGseq/man/DEGseq.pdf, accessed on 19 November 2023). For the examination of transcriptional gene set enrichment, the GSEA KEGG method with ClusterProfiler version 4.2.2 [48] was applied. We also conducted protein–protein interaction analysis using STRING version 11.5 (https://string-db.org/, accessed on 19 November 2023).

### 4.9. Statistics and Data Analysis

All the experimental values were presented as the mean ± SEM (standard error of the mean). The analysis was performed using GraphPad Prism version 9. Statistical analysis was performed using the Student’s *t*-test. A *p*-value of less than 0.05 was considered to be statistically significant.

## 5. Conclusions

Our findings showed that FT895 damaged MPNST cells by compromising mitochondrial functions, which resulted in increased ROS levels, a reduction in mitochondrial DNA copy numbers, and impairment of mitochondrial respiration. The mechanism of mitochondrial dysfunction might result from the down-regulation of XBP1s, which controls the biogenesis and mitophagy of mitochondria at the transcriptional level. The RNA-seq analysis outlined a mechanism for the mitochondrial dysfunction post-FT895 treatment, potentially involving the HIF-1α signaling pathway coupled with the dysregulation of glucose-energy production. These findings highlight the potential for a combined therapeutic approach utilizing FT895 alongside other drugs with different mechanisms to combat MPNST.

## Figures and Tables

**Figure 1 ijms-25-00277-f001:**
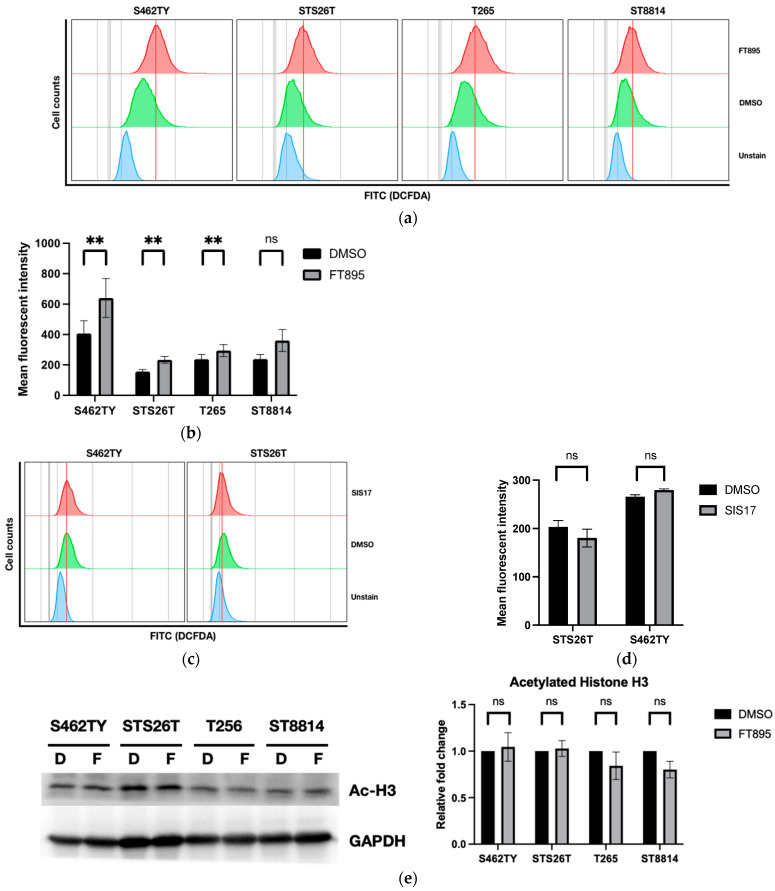
Schemes follow the same formatting. The increase in reactive oxygen species (ROS) levels in MPNST cells after FT895 treatment. (**a**) FACS graphs with fluorescent DCFDA cell counts in MPNST cells treated with 10 μM FT895 for 24 h. (**b**) The mean fluorescent intensities of the DCFDA signals in cells treated with either DMSO (control) or FT895 were shown as mean ± SEM. (**c**) FACS graphs demonstrate the cell counts with fluorescent DCFDA in S462TY and STS26 cell lines treated with 10 μM SIS17 for 24 h. (**d**) The mean fluorescent intensities of DCFDA signals in S462TY and STS26T cells treated with DMSO (control) or SIS17. (**e**) Quantification of acetylated histone H3 in MPNST cells after treatment of FT895 (**f**) The relative fold changes of the DNA copy numbers of ND2 and ND4L genes were calculated based on quantitative PCR. Data were calculated from three independent experiments. Statistical analysis was performed by unpaired, two-tailed Student’s *t*-test (* represents *p* ≤ 0.05, ** represents *p* ≤ 0.01, *** represents *p* ≤ 0.001, ns represents not significant).

**Figure 2 ijms-25-00277-f002:**
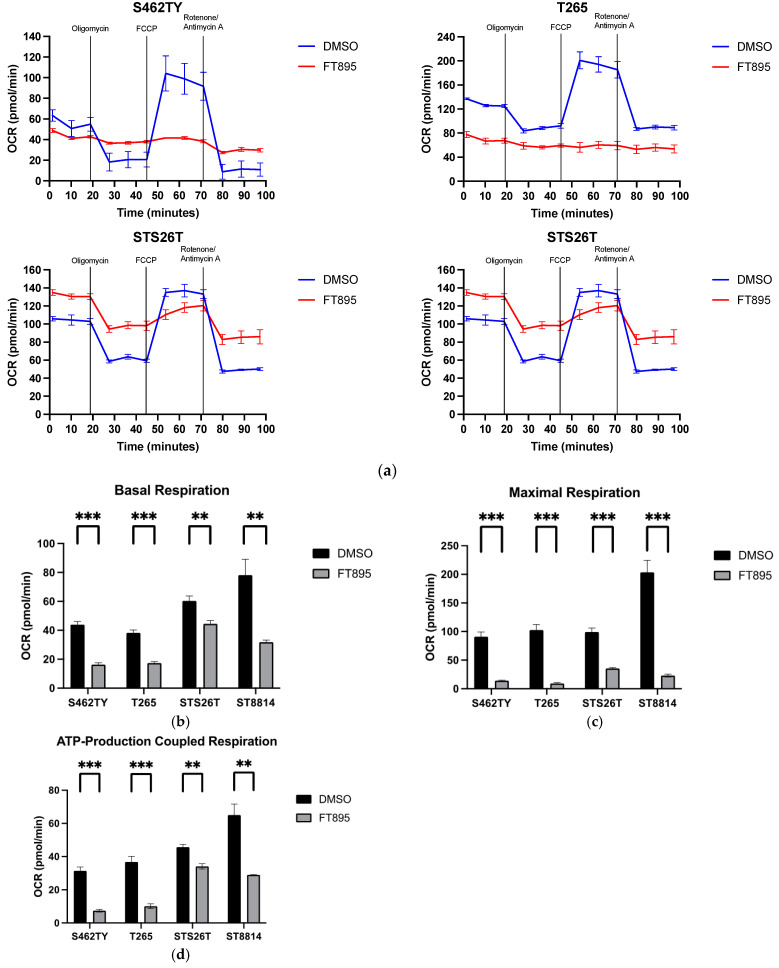
Mitochondrial respiration is affected by FT895-treated MPNST cells. Cell lines, S462TY, T265, STS26T, and ST8814 were treated with 10 µM FT895 for 24 h. The mitochondrial respiration of four MPNST cells was measured by injection of oligomycin, FCCP, and rotenone/antimycin and plotted as the kinetic graphs (**a**). The basal (**b**), maximal (**c**), and ATP-production-coupled respiration (**d**) were analyzed from the kinetic data. (**e**) The energy map for the MPNST cells after treatment indicates the shifting of phenotypes driven by FT895. The kinetic graphs (**f**) of mitochondrial respiration for S462TY and T265 treated with 10 and 20 μM FT895 for 6 h. Following a 6 h treatment with FT895, both basal (**g**) and maximal respiration (**h**) of mitochondria in S462TY and T265 cells exhibited a dose-dependent reduction when compared to those treated with DMSO. Data were calculated from three independent experiments. Statistical analysis was performed by unpaired, two-tailed Student’s *t*-test (** represents *p* ≤ 0.01, *** represents *p* ≤ 0.001, ns represents not significant). Raw data were analyzed using Seahorse Wave Pro software prior (https://www.agilent.com.cn/zh-cn/product/cell-analysis/real-time-cell-metabolic-analysis/xf-software/software-download-for-seahorse-wave-pro-software, accessed on 19 December 2023) to the graphical presentation using GraphPad Prism version 9.5.0.

**Figure 3 ijms-25-00277-f003:**
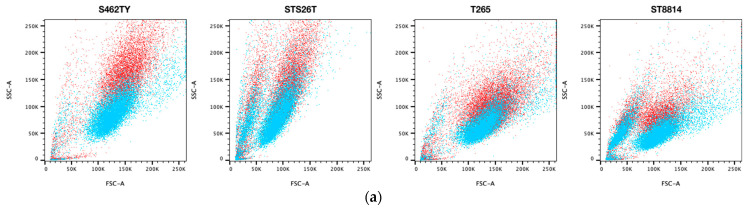
Increased granularity and mitochondria aggregates in MPNST cells after 10 µM FT895 treatment. (**a**) FACS graphs of forward scatter (FSC) and side scatter (SSC). The blue color indicates the cells treated with DMSO, while red is the FT895-treated group. (**b**) Co-localization of mitochondria (green), autophagosome (red), and nucleus (blue) in S462TY treated with 10 µM FT895 for 24 h. Data were obtained from three independent experiments.

**Figure 4 ijms-25-00277-f004:**
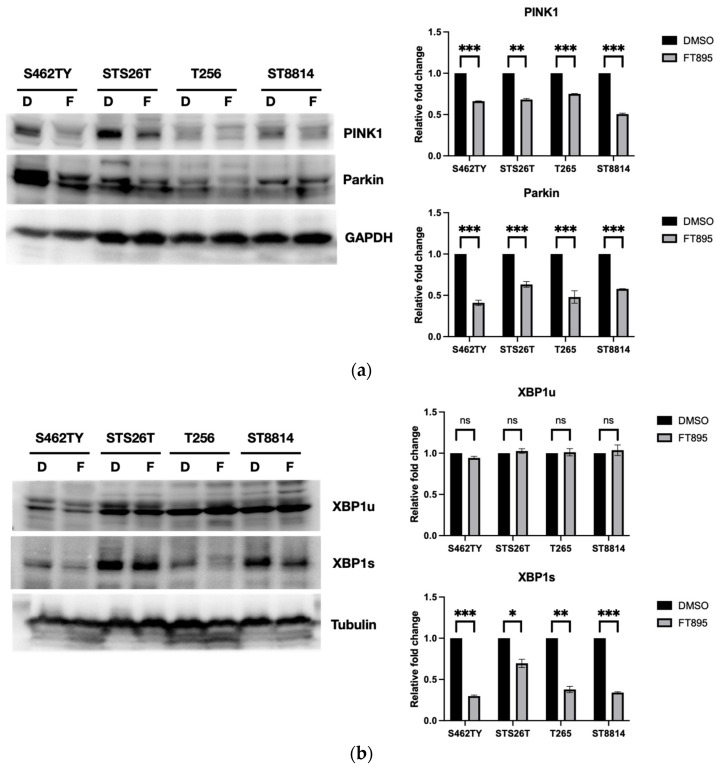
Western blotting to show the impact of FT895 on PINK1, Parkin, and XBP1 protein levels. MPNST cells were treated with DMSO (D) or 10 µM FT895 (F) for 24 h. The expression of PINK1 and Parkin proteins (**a**). The unspliced XBP1 (XBP1u) and spliced XBP1 (XBP1s) protein expression (**b**). Data were obtained from three independent experiments (* represents *p* ≤ 0.05, ** represents *p* ≤ 0.01, *** represents *p* ≤ 0.001, ns represents not significant).

**Figure 5 ijms-25-00277-f005:**
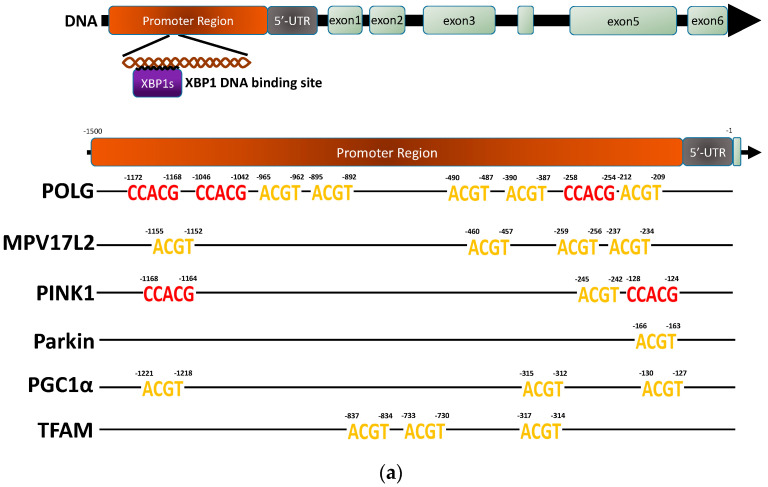
XBP1s modulated the transcriptional activity of genes associated with mitochondria in MPNST cells treated with FT895. XBP1 protein is a transcription factor that mediates the transcription of the candidate genes, *POLG*, *MPV17L2*, *PINK1*, *Parkin*, *PGC1α*, and *TFAM*. The sequence of its binding sites in the promoter region was shown (**a**). The promoter DNA fragments of the candidate genes were immunoprecipitated by anti-XBP1s antibody. The changes in the DNA copy numbers of these promoter fragments were semi-quantified via real-time PCR experiments (**b**). The mRNA expression levels of candidate genes were semi-quantified by real-time PCR (**c**). Data were calculated from three independent experiments. Statistical analysis was performed by unpaired, two-tailed Student’s *t*-test (* represents *p* ≤ 0.05, ** represents *p* ≤ 0.01, *** represents *p* ≤ 0.001, ns represents not significant).

**Figure 6 ijms-25-00277-f006:**
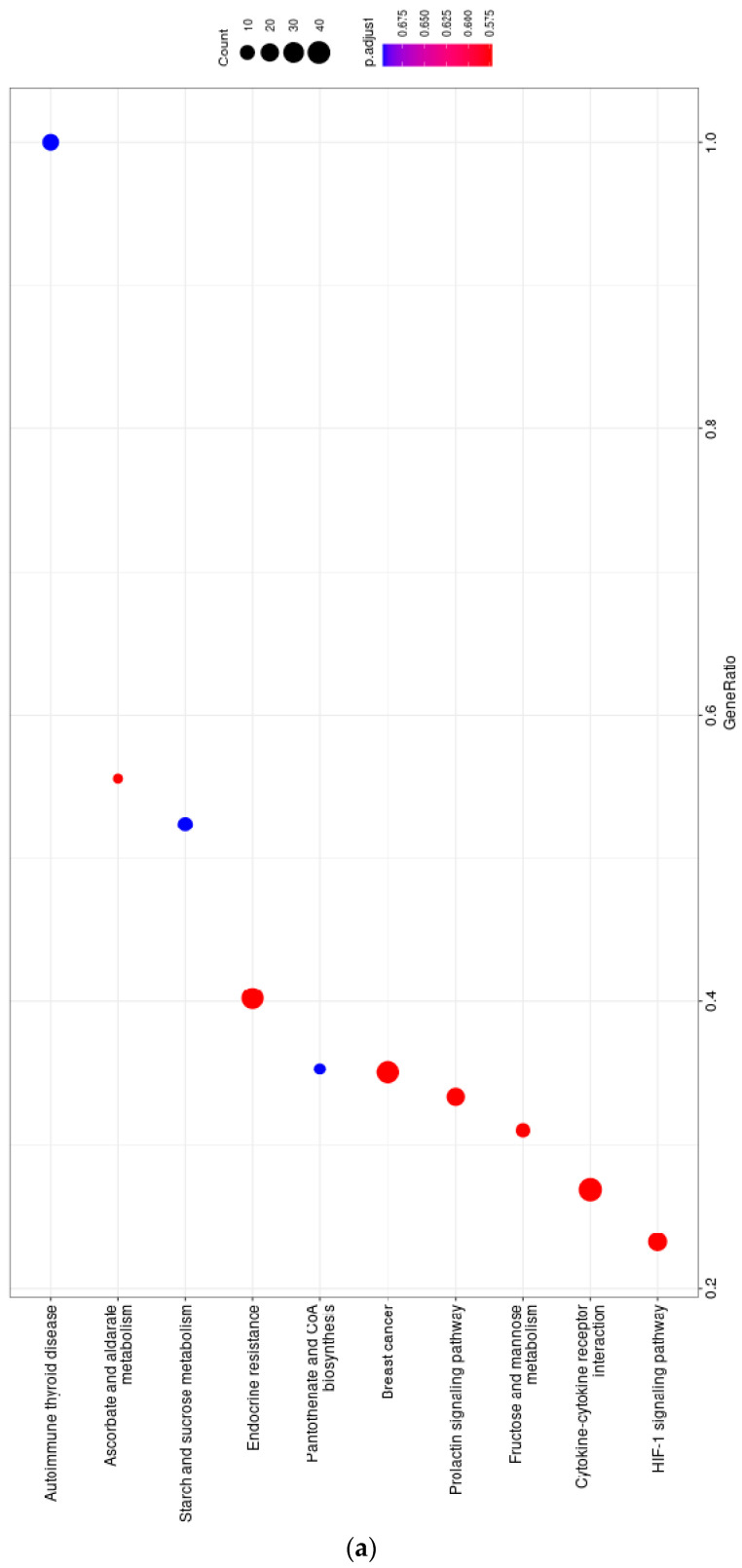
Pathway analysis of RNA-seq in S462TY with 6 h treatment of 10 µM FT895. Dot plot of the top 10 KEGG pathway (**a**). The protein–protein interaction network is depicted by the STRING database and web source (**b**).

**Figure 7 ijms-25-00277-f007:**
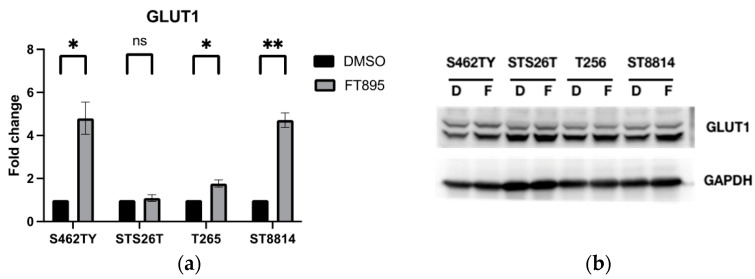
GLUT1 expression in MPNST cells treated with 10 µM FT895 for 24 h. The fold changes in mRNA level were quantified by real-time PCR (**a**), and protein levels were detected by Western blot (**b**). Data were obtained from three independent experiments (* *p* < 0.05, ** *p* < 0.01, ns represents not significant).

**Figure 8 ijms-25-00277-f008:**
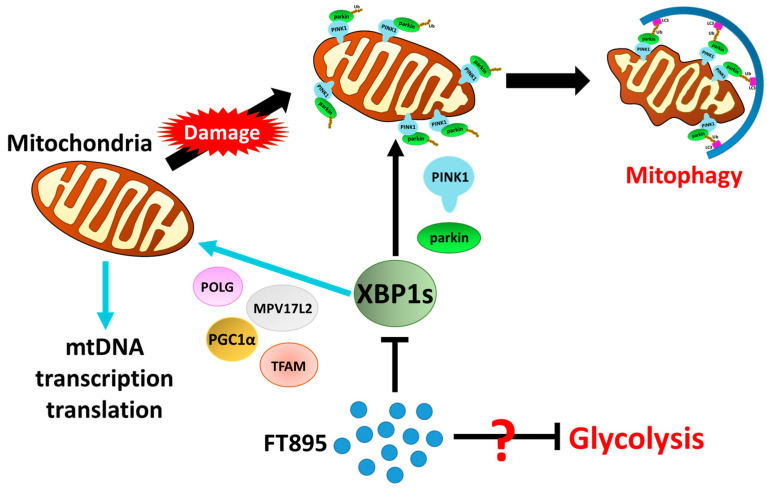
Propose molecular mechanism of FT895-induced mitochondrial damage and deficient mitophagy.

**Table 1 ijms-25-00277-t001:** Pathways associated with FT895-affected genes according to gene set enrichment analysis based on KEGG.

ID	Pathway	Set Size	Enrichment Score	*p* Value	*p*. Adjust	Core Enrichment	Up-/Down-Regulation
hsa00051	Fructose and mannose metabolism	29	0.676224664	0.00175	0.57248645	PFKFB4/ALDOC/HK2/ALDOB/HK1/ALDOA/PFKFB3/PFKL/MPI	Up
hsa00053	Ascorbate and aldarate metabolism	9	-0.814634067	0.009477	0.57248645	ALDH2/AKR1A1/ALDH3A2/ALDH1B1/MIOX	Down
hsa04917	Prolactin signaling pathway	60	0.508891224	0.011784	0.57248645	FOS/STAT5A/PIK3R2/IRF1/MAP2K1/FOXO3/ESR2/RELA/STAT3/LHB/MAPK13/TNFRSF11A/TNFSF11/PIK3CD/PIK3CA/SOS1/PIK3R1/SRC/AKT3/AKT3	Up
hsa04066	HIF-1 signaling pathway	99	0.468791063	0.005213	0.57248645	ENO2/PDK1/ALDOC/PIK3R2/SLC2A1/HK2/LDHAL6A/ALDOB/CAMK2B/EGLN1/HK1/CDKN1A/PGK1/MAP2K1/CYBB/VEGFA/LDHA/LDHA/EDN1/ALDOA/PFKFB3/INSR/PFKL	Up
hsa01522	Endocrine resistance	92	0.456475643	0.011786	0.57248645	FOS/PIK3R2/JUN/JAG2/CDKN1A/MAP2K1/ADCY5/NOTCH4/NOTCH4/NOTCH4/NOTCH4/NOTCH4/NOTCH4/NOTCH4/DLL1/DLL1/E2F2/E2F2/HBEGF/ESR2/JAG1/RB1/E2F1/NOTCH1/MAPK13/ADCY1/PIK3CD/IGF1R/ADCY6/PIK3CA/ADCY8/SOS1/PIK3R1/SRC/AKT3/AKT3/BRAF	Up
hsa01522	Endocrine resistance	92	0.456475643	0.011786	0.57248645	FOS/PIK3R2/JUN/JAG2/CDKN1A/MAP2K1/ADCY5/NOTCH4/NOTCH4/NOTCH4/NOTCH4/NOTCH4/NOTCH4/NOTCH4/DLL1/DLL1/E2F2/E2F2/HBEGF/ESR2/JAG1/RB1/E2F1/NOTCH1/MAPK13/ADCY1/PIK3CD/IGF1R/ADCY6/PIK3CA/ADCY8/SOS1/PIK3R1/SRC/AKT3/AKT3/BRAF	Down
hsa05224	Breast cancer	134	0.419091155	0.011352	0.57248645	FOS/PIK3R2/GADD45G/JUN/JAG2/FGF10/WNT9A/WNT7B/CDKN1A/FGF20/MAP2K1/NOTCH4/NOTCH4/NOTCH4/NOTCH4/NOTCH4/NOTCH4/NOTCH4/FGF7/WNT1/NFKB2/DLL1/DLL1/WNT10A/FZD1/E2F2/E2F2/ESR2/TCF7L1/JAG1/FGF1/FGF22/PGR/RB1/WNT3/WNT3/WNT3/E2F1/FZD4/FRAT1/KIT/EGF/NOTCH1/FLT4/WNT11/FZD3/FGF17	Up
hsa00500	Starch and sucrose metabolism	21	0.656614559	0.016998	0.69880852	HK2/HK1/PGM1/GYS1/GBE1/GPI/GPI/PGM2/PYGM/UGP2/PYGL	Up
hsa05320	Autoimmune thyroid disease	15	-0.681660688	0.032885	0.69880852	HLA-DQB1/HLA-DQB1/HLA-DQB1/HLA-DQB1/HLA-DQB1/HLA-DQB1/HLA-DQB1/HLA-DOB/HLA-DOB/HLA-DOB/HLA-DOB/HLA-DOB/HLA-DOB/PRF1/TSHR	Down
hsa00770	Pantothenate and CoA biosynthesis	17	-0.66148836	0.032462	0.69880852	ALDH3A2/ALDH1B1/BCAT2/COASY/UPB1/DPYS	Down

## Data Availability

We would provide the details what the readers are interested in.

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
