# Peer review of "FT895 Impairs Mitochondrial Function in Malignant Peripheral Nerve Sheath Tumor Cells"

_ijms, 2023, doi:10.3390/ijms25010277_

Round 1
Reviewer 1 Report
Comments and Suggestions for Authors
Review comment,
In the present manuscript, it indicated that FT895, an HDAC11 inhibitor, disrupts mitochondrial biogenesis and function in the malignant peripheral nerve sheath tumor (MPNST) cell lines due to reduction of the spliced XBP1 (XBP1s) which regulates the mitochondria-related genes (MPV17L2, POLG, TFAM, PGC1a, PLINK and Parkin). However, the manuscript is not well-written. I recommend that this paper accepted after major revision.
1. It is not unclear the mechanism that how FT895 directly affect the mitochondrial biogenesis and function in MPNST cell lines. FT895 affects the mitochondrial function. On the other hand, SIS17, an HDAC11 inhibitor did not affect the mitochondrial function. Does FT895 affect the mitochondrial function due to non-specific effect except the inhibition of HDAC11? It had better to show the quantification of acetylated histone.
2. The XBP1s enrichment of the promoter region in mitochondria-related genes such as PINK1 and Parkin were shown in Figure 5b. However, the expression levels of these candidate genes are not shown. In ST8814 cell line, the no difference of enrichment was shown in Figure 5b. Does the expression level of candidate genes change in ST8814 cell line?
3. It is difficult to understand the effect of the reduction of mitochondrial function and glycolytic rates between the MPNST cell lines (such as the NF1-derived and non-derived). It should show the cell growth rate under all conditions.
Reviewer 2 Report
Comments and Suggestions for Authors
The paper describes a FT895 dependent impairment of mitochondrial function in MPNST cells. I have the following comments.
1. As shown in Fig. 1 a and f mitochondrial impairment by 10µM FT895 is dramatically time dependent, but only little concentration dependent. That time dependency needs to be investigated in greater detail. What about cellular viability after 24h?
2. In this respect it is unclear to me why RNAseq was performed after 6h but the GLUT1 expression changes were investigated after 24h.
3. It is unclear why in the RNAseq data no mtDNA related transcripts are downregulated, given the strong effects on mtDNA copy number.
4. Fig.6a is not very informative. Please present a table with significant up- and downregulated pathways.
5. The quality of WB's presented in Fig. 4 is very poor. Please present better quality blots and bar graphs with error bars.
6. In the legends to the figures the numbers of performed experiments are not indicated.
7. The legend to Fig. 1 needs to be corrected (remove the phrases from the template).
Comments on the Quality of English Language
None.
Round 2
Reviewer 1 Report
Comments and Suggestions for Authors
In the present manuscript, it indicated that FT895, an HDAC11 inhibitor, disrupts mitochondrial biogenesis and function in the malignant peripheral nerve sheath tumor (MPNST) cell lines due to reduction of the spliced XBP1 (XBP1s) which regulates the mitochondria-related genes (MPV17L2, POLG, TFAM, PGC1a, PLINK and Parkin). Although the cover letter includes responses to the previous review comments, the manuscript does not include any responses. I recommend that this paper accepted after minor revision.
Reviewer 2 Report
Comments and Suggestions for Authors
The author have addressed my concerns accordingly.
Comments on the Quality of English LanguageNone.
Author Response
We sincerely thank you for the valuable comments.